# Antibacterial Effect and Possible Mechanism of Sesamol against Foodborne Pathogens

**DOI:** 10.3390/foods13030435

**Published:** 2024-01-29

**Authors:** Zhuosi Li, Mengjie Wu, Hui Yan, Zheyan Meng, Binru Gao, Qingli Dong

**Affiliations:** School of Health Science and Engineering, University of Shanghai for Science and Technology, Shanghai 200093, China; lizhuosi@usst.edu.cn (Z.L.); wumengjie202102@163.com (M.W.); 15849338401@163.com (H.Y.); janemeng0604@163.com (Z.M.); gao_binru@163.com (B.G.)

**Keywords:** sesamol, foodborne pathogens, antibacterial effect, mechanism

## Abstract

Food safety problems caused by foodborne pathogens have become a major public issue, and the search for efficient and safe bacteriostatic agents has gained attention. Sesamol (SE), a phenolic compound abundant in sesame oil, offers numerous health benefits and exhibits certain antibacterial properties. The purpose of this study was to evaluate the antibacterial effect and potential mechanisms of SE against representative foodborne pathogens, including *Listeria monocytogenes*, *Staphylococcus aureus*, *Bacillus cereus*, *Escherichia coli*, and *Salmonella* serovar Enteritidis. The results showed that SE significantly inhibited the growth of the five pathogenic bacteria in sterile saline and pasteurized milk by 2.16–4.16 log_10_ CFU/g within 48 h. The results of the minimum bactericidal concentration and time–kill assay showed that SE had a greater inhibitory effect on *L. monocytogenes* compared with other bacteria. Additionally, SE was found to alter the cell membranes’ permeability in these bacteria, resulting in the release of intercellular proteins and DNA. A scanning electron microscopy analysis showed that exposure to SE resulted in significant changes in bacterial morphology, producing cell shrinkage and deformation. These findings suggest that SE could inhibit both Gram-negative and Gram-positive bacteria by interfering with the function and morphology of bacterial cells.

## 1. Introduction

Foodborne pathogens are a group of pathogenic bacteria that spread and cause food poisoning through contaminated food [1]. *Listeria monocytogenes*, an invasive Gram-positive bacterium, has recently been frequently detected in enoki mushrooms, ready-to-eat foods, and raw meat products [2]. It can cause serious invasive listeriosis, with a fatality rate as high as 20–30% [2]. *Staphylococcus aureus*, a Gram-positive foodborne pathogen, can secrete heat-stable enterotoxins, which cause staphylococcal food poisoning [3]. *Bacillus cereus*, a Gram-positive bacterium, can generate a ceruleite toxin that induces vomiting and toxins that cause diarrhea [4,5]. *Salmonella* serovar Enteritidis, a Gram-negative bacterium, is frequently responsible for non-typhoidal salmonellosis in humans and is mainly isolated from eggs and poultry meat [6,7]. *Escherichia coli*, a bacterium classified as Gram-negative, continues to be a prevalent source of various bacterial infections in both humans and animals [8]. These representative bacteria infecting humans may lead to various diseases such as gastrointestinal inflammation, meningitis, suppurative lymphadenitis, and even death [9]. Therefore, it is necessary to employ biological, physical, or chemical methods to reduce microbial contamination in food and effectively decrease the occurrence of foodborne illnesses.

Due to consumers’ concerns about the safety of artificial additives used in food preservation, as well as their growing focus on health, natural bacteriostatic agents extracted from plants, animals, bacteria, algae, and fungi have attracted attention from both consumers and researchers [10,11]. Phenolic compounds, found in plants as secondary metabolites, possess significant value as natural molecules due to their bioactive characteristics, such as antioxidant properties [12]. Some plant phenolics or phenolic-rich extracts exhibit remarkable inhibitory effects on a broad spectrum of bacteria [12]. Phenolic compounds have the potential to induce alterations in bacterial morphology, disrupt the integrity of bacterial cell walls, and affect the formation of biofilms [11,12]. They additionally exert an impact on the biosynthesis of proteins, the metabolic processes of cells, and the synthesis of ATP and DNA [13]. The hydroxyl groups present in phenols can reduce the catalytic capacity of enzymes by forming hydrogen bonds, and they have also been reported to alter the proton motive force both inside and outside the cell [14,15]. Hence, the utilization of plant phenolic compounds presents a promising alternative approach to the application of chemical preservatives within the food sector.

Sesamol (SE, 3,4-methedioxyphenol), a phenolic compound found in sesame seeds, sesame oil, and sesame meal, exhibits antimicrobial, anticancer, antiangiogenic, immunomodulatory, cardioprotective, and antioxidant activities [16,17,18]. In our previous research, SE exhibited an inhibitory effect on the growth of *L. monocytogenes* strains isolated from food or clinical samples [19,20]. SE also showed potential for reducing *L. monocytogenes* contamination in a chilled tuna sample [20]. Furthermore, SE has been reported to inhibit the growth of both Gram-positive bacteria (*S. aureus* and *L. monocytogenes*) and Gram-negative bacteria (*E. coli* and *Salmonella enterica* serovar Typhimurium) [21]. A further advantage of SE is that it lacks the odors or taste associated with other antibacterial phenols such as carvacrol, coumarin, gallic acid, and catechin, making it more appealing for use [12,22]. Our previous results showed that the addition of SE (1.0 mg/mL) in frozen yellowfin tuna fillets did not cause organoleptic alteration, including the texture, color characteristics, and sensory scores of the tuna fillets [20]. The use of SE (0.5 g/kg) was reported to have not altered the organoleptic properties of meatballs [21].

However, there is still a lack of comprehensive comparison and analysis regarding the antibacterial effect and possible mechanism of SE against various foodborne pathogens. In particular, it remains unclear whether the disruption of the bacterial cell membrane and the release of intracellular substances are key factors that contribute to bacterial inactivation.

Therefore, in this study, the impact of SE on five typical foodborne pathogens was investigated in both a sterile saline solution and pasteurized milk to broaden the potential applications of SE in different food environments. To elucidate the possible mechanisms of SE, the permeability of bacterial cell membranes, the leakage of cell contents, and the changes in cell morphology were evaluated. These data could provide scientific evidence supporting the effectiveness of SE as a natural antimicrobial agent in food.

## 2. Materials and Methods

### 2.1. Chemical Reagents and Bacterial Strains

SE (CAS: 533-31-3, 98%) was obtained from Leyan (Shanghai, China). SE was resuspended in trypticase soy-yeast extract broth (TSB-YE) with 0.10% dimethylsulfoxide. The *L. monocytogenes* ATCC 19112 strain, the *B. cereus* ATCC 14579 strain, the *S. aureus* ATCC 29213 strain, and the *E. coli* ATCC 25922 strain were obtained from the China Center of Industrial Culture Collection. *S.* Enteritidis MRL 04120 was isolated from food samples in our laboratory (Table 1).

### 2.2. Bacterial Culture

All strains were stored in brain–heart infusion broth (BHI; Beijing Luqiao Co., Beijing, China) containing 50% (*v*/*v*) glycerol at −80 °C. Working stocks of all strains were stored at 4 °C on tryptone soy agar (TSA; Beijing Land Bridge Technology Co., Ltd., Beijing, China) and renewed monthly. Single colonies of each bacterium were selected and cultured in 10 mL of BHI at 37 °C for 12–24 h. Subsequently, the strain cultures were centrifuged at 21,127× *g* for 10 min (at 4 °C), washed three times, and then resuspended in 0.85% sterile saline solution (SSS, pH = 7.2) to obtain inoculums of 10^8^–10^9^ CFU/mL.

### 2.3. Determination of Minimum Inhibitory Concentration (MIC) and Minimum Bactericidal Concentration (MBC)

The MICs of SE against five bacterial strains were determined using the broth microdilution method, as previously reported [14]. Briefly, the aforementioned bacteria were diluted in SSS to achieve a final concentration of 10^6^ CFU/mL. Then, 100 uL of the bacterial suspension and 100 μL of SSS or BHI containing SE (final concentration: 3, 2.5, 2, 1.5, or 0 mg/mL) were added to the 96-well plate, respectively. After incubation at 37 °C for 24 h, the OD_600 nm_ was measured using a microplate reader (Read-Max1900, Shanghai Shanpu Biotechnology Co., Ltd.). The MIC was defined as the minimum concentration of SE at which no bacterial growth was observed. When reading the MIC results, 100 μL of each dilution mixture that exceeded the MIC was inoculated onto agar plates and incubated at 37 °C for 24 h. The MBC is determined as the concentration at which the colony count on the agar plate is less than 5. Penicillin K was used as a positive control.

### 2.4. Time–Kill Assays in SSS and Pasteurized Milk

The time–kill assay was performed in accordance with a previous report [23]. Commercial pasteurized milk, or SSS, was inoculated with five strains, each at a concentration of 10^6^ CFU/mL, respectively. SSS, or milk containing only bacteria, was used as the control. The samples, treated with or without SE (1/2× MIC and 1× MIC), were incubated at 37 °C and sampled at predetermined time points (0, 6, 12, 24, and 48 h) for plate counting.

### 2.5. Electrical Conductivity (EC) Measurement

The cell membrane permeability of the five strains was examined by measuring EC [24]. Briefly, bacterial suspensions diluted with SSS (10^8^–10^9^ CFU/mL) were treated with or without SE (1/2× MIC and 1× MIC), incubated at 37 °C, and sampled at predetermined time points (0, 12, 24, and 48 h). SSS was used as a control. The samples (2.0 mL) were centrifuged at 6952× *g* for 10 min. The supernatant was used to determine EC using a conductivity meter (INASE Scientific Instrument Co., Ltd., Shanghai, China).

### 2.6. Determination of Deoxyribonucleic Acid (DNA) Leakage

All bacteria were treated as described in Section 2.5 above, and simple modifications were made based on previous methods [25]. After centrifugation, 200 μL of the supernatant was transferred to a 96-well plate. The DNA content in the supernatant was quantified by measuring the absorbance at 260 nm using a microplate reader (Read-Max1900, Shanpu Biotechnology Co., Ltd., Shanghai, China).

### 2.7. Bicinchoninic Acid (BCA) Protein Assay

All bacteria were treated as described in Section 2.5 above. After centrifugation, the supernatant was utilized for protein concentration measurement using a BCA Protein Assay Kit (Beyotime Biotechnology Co., Ltd., Shanghai, China).

### 2.8. Determination of ATP Concentration

All bacteria were treated as described in Section 2.5 above. Following centrifugation, the supernatant was utilized for the ATP concentration determination using an ATP assay kit (Enzyme Link Biotechnology Co., Ltd., Shanghai, China). The absorbance at 450 nm was measured using a microplate reader (Read-Max1900, Shanpu Biotechnology Co., Ltd., Shanghai, China).

### 2.9. Scanning Electron Microscope (SEM) Analysis

Bacterial suspensions diluted with 2.0 mL of SSS (10^8^–10^9^ CFU/mL) were treated with or without SE (1× MIC and 2× MIC) and incubated at 37 °C for 24 h. The samples were centrifuged at 21,127× *g* for 10 min to remove the supernatant, followed by PBS washing and overnight fixation at 4 °C using a 2.5% glutaraldehyde solution. After removing the glutaraldehyde, the samples were rinsed three times with PBS. The samples were dehydrated using different concentrations of ethanol (30%, 50%, 70%, 90%, and twice with 100%) for 10 min each. Subsequently, they were sputter-coated with gold and visualized using SEM (Tescan Mira 3 XH).

### 2.10. Statistical Analysis

The data were presented as mean values ± standard deviations (SD) (n = 3). Statistical significance of differences was assessed using analysis of variance (ANOVA) and Duncan’s multiple range test, which were performed with SPSS 21.0 software (SPSS Inc., IBM Corporation, Armonk, NY, USA). The significance of the differences was assessed at a 95% level of confidence (*p* < 0.05). All experiments were conducted in triplicate.

## 3. Results and Discussion

### 3.1. MIC and MBC of SE against Five Bacterial Strains

The MIC and MBC of SE were investigated for five bacterial strains, and the results are shown in Table 1. SE showed antimicrobial activity against five pathogenic bacteria, with MICs ranging from 1.5 to 2.0 mg/mL and MBCs ranging from 4.0 to 16.0 mg/mL. There was no significant difference between the MIC values obtained in SSS and the BHI medium. It was observed that the MICs of SE were a bit lower for two Gram-negative strains (*E. coli* and *S.* Enteritidis) at 1.5 mg/mL compared with two Gram-positive bacteria (*B. cereus* and *L. monocytogenes*) at 2.0 mg/mL. However, the MIC of Gram-positive *S. aureus* was 1.5 mg/mL. The difference in architecture and molecular components of the cell peripheral wall between Gram-positive and Gram-negative bacteria may contribute to the different mechanisms of antimicrobial agents [26]. However, this study did not find a significant difference in the effect of SE between Gram-positive and Gram-negative bacteria. In our previous study, the MIC of SE against *L. monocytogenes* (ATCC19112, food-derived strains, and clinical stains) was found to be 1.5–2.0 mg/mL in TSB-YE [19]. The MIC of SE was 2 mg/mL against *B. cereus* (F4810), and *S. aureus* (FRI 722) [27]. In this study, the inhibitory effect of SE on pathogenic bacteria was investigated in SSS, which is less nutrient-rich than standard medium and therefore able to simulate the environment of some actual food. The MIC of *L. monocytogenes* ATCC 19112 in SSS was consistent with the results in the TSB-YE medium [19]. The results of MBCs showed that *L. monocytogenes*, *B. cereus,* and *E. coli* were more sensitive to SE, while *S. aureus* was difficult to inhibit. These results show that the inhibitory effect of SE may be affected by the matrix and strains. In the future, the effect of SE needs to be evaluated on multiple strains (different types and sources) as well as food matrices (nutrient content and pH value).

The MIC of penicillin G (a standard antibiotic) against different *S. aureus* strains ranged from 0.003 to 20 μg/mL [28]. In comparison to the standard antibiotic, SE had a high MIC. To confirm the bacteriostatic activity of SE, the MIC values of other phenolic substances with antimicrobial activity were compared. For example, the MIC of thymol against *L. monocytogenes* was 2 mg/mL [29]. The coumarin against *E. coli* and *S.* Typhimurium was 0.625–5 mg/mL; however, the MIC of 11 phenolic compounds against various foodborne bacteria, such as *L. monocytogenes, S. aureus, E. coli,* and *S*. Typhimurium, ranged from 125 to ≥500 µg/mL [30]. Although SE alone exhibits a high MIC, we were pleasantly surprised to discover that when combined with other antimicrobial substances such as nisin, the MICs against *L. monocytogenes* can be reduced to 0.375–0.750 mg/mL [19]. Therefore, SE may be more suitable as an auxiliary bacteriostatic than a single one.

The consumption of SE has been established as safe. According to reports, the estimated daily intake (EDI) of SE from sesame seeds and sesame oil is 0.11 mg/person/day [31]. In animal experiments investigating the physiological activities of SE, such as its anticancer effects in mice or rats, the typical dosage ranges from 1 to 100 mg/kg [32]. Extrapolating this dosage to humans would be equivalent to 60–6000 mg for an adult weighing 60 kg. The dosage of 1000 mg obtained from consuming 500 mL of milk, which is based on the MIC of SE at a concentration of 2 mg/mL in milk, is significantly lower than the typical dosage ranges mentioned above. In addition, the toxicity of SE for normal cells (mouse epithelial fibroblast cells L929) was investigated. SE at 1/2× MIC and 1× MIC reduced the viability of the cells by 5–10%. Therefore, the use of SE as a food additive might be considered safe.

### 3.2. Inhibitory Effect of SE on the Growth of Five Bacterial Strains in SSS and Pasteurized Milk

To gain a deeper understanding of the impact of SE in a realistic food environment, time–kill assays were conducted on five bacterial strains in both SSS and pasteurized milk. From 0 h to 48 h, the concentration of *L. monocytogenes* in SSS increased from 6.0 log_10_ CFU/mL to 7.09 log_10_ CFU/mL (Figure 1A). Compared with the control group, both SE at 1/2× MIC and 1× MIC exhibited significant inhibitory effects on the growth of *L. monocytogenes* in SSS (Figure 1A). At 24 h and 48 h, the concentration of *L. monocytogenes* in SSS was significantly reduced by 3.63 log_10_ CFU/mL and 2.99 log_10_ CFU/mL, respectively, when treated with 1× MIC of SE (Figure 1A). The milk, compared with the less nutrient-rich SSS, significantly promotes the rapid growth of *L. monocytogenes*, resulting in an increase to 8.38–8.72 log_10_ CFU/mL within 12–48 h (Figure 1B). Both 1/2× MIC and 1× MIC of SE showed a significant inhibitory effect on *L. monocytogenes* in milk within 48 h compared with the control group (Figure 1B). At the end of the 48 h period, both 1/2× MIC and 1× MIC of SE demonstrated optimal inhibitory effects, reducing the concentration of *L. monocytogenes* by 3.29 and 4.16 log_10_ CFU/mL, respectively (Figure 1B).

In our previous study, after 48 h of treatment with a 1× MIC concentration of SE, the population of *L. monocytogenes* ATCC 19112 strains in TSB-YE and pasteurized milk was reduced by 4.35 and 5.25 log_10_ CFU/mL, respectively [19]. These results indicate that, compared with the less nutrient-rich SSS, the effect of SE against *L. monocytogenes* in nutrient-rich milk, or TSB-YE, might be more effective. Our previous study showed that SE (1× MIC) significantly decreased the concentration of *L. monocytogenes* in chilled raw tuna fillets by 1.29–2.65 log_10_ CFU/g at 10 °C after 72 h [20]. Reportedly, at days 9 and 12, and at 3 ± 1 °C, the *L. monocytogenes* (ATCC 51774) count in meatballs treated with SE at 0.5 and 0.3 g/kg concentrations was completely reduced (ranging from 7.67 to 8.28 log_10_CFU/g on day 0) [21]. It is worth noting that although the inhibition results of SE on *L. monocytogenes* in meatballs [21] and tuna [20] were used for comparison, SE exhibits different diffusivity, solubility, and persistence in solid food, viscoelastic food, and liquid food. Therefore, future studies should investigate the antibacterial effects of SE in different food matrices. From 0 h to 48 h, the concentration of *S. aureus* in SSS increased from 6.0 log_10_ CFU/mL to 6.99 log_10_ CFU/mL (Figure 1C). Both 1/2× MIC and 1× MIC of SE showed a significantly inhibitory effect on the growth of *S. aureus* in SSS at all times within 48 h compared with the control group (Figure 1C). In SSS, the SE at 1× MIC exhibited the highest reduction in *S. aureus* at 12 h compared with other time points, with a decrease of 3.29 log_10_ CFU/mL (Figure 1C). Similar to *L. monocytogenes*, the growth of *S. aureus* in milk was faster, reaching 8.89 log_10_ CFU/mL at the end of the 48 h period (Figure 1D). The decrease in *S. aureus* in both milk and SSS exhibited a positive correlation with the dosage of SE. Compared with the control group, SE at 1× MIC significantly reduced the concentration of *S. aureus* in milk by 2.62, 2.50, and 2.30 log_10_ CFU/mL at 12, 24, and 48 h, respectively (Figure 1D).

The antimicrobial effect of sesame oil, or SE, against *S. aureus* in food products has been studied previously. Meatballs, when treated with different amounts of SE (0.3 and 0.5 g/kg), exhibited a significant reduction in *S. aureus* counts by approximately 4.1 and 5.1 log_10_ CFU/g after being stored for 3 days at a temperature of approximately 3 ± 1 °C [21]. Additionally, sesame oil demonstrated a significant antibacterial effect against *S. aureus*; however, the specific active substance responsible for this effect remains unclear due to the presence of various compounds in sesame oil, such as sesamin, SE, and phenolic volatile oil (eugenol) [33,34].

The concentration of *B. cereus* in SSS initially increased to a maximum of 7.64 log_10_ CFU/mL at 24 h and then slightly decreased thereafter (Figure 1E). At 24 and 48 h, the inhibitory effect of SE at 1× MIC on *B. cereus* in SSS was more significant than that at 1/2× MIC (Figure 1E). At 24 h, the reduction in *B. cereus* in SSS caused by SE at 1× MIC was most significant, with a decrease of 2.79 log_10_ CFU/mL compared with the control group (Figure 1E). Similar to *L. monocytogenes* and *S. aureus*, the growth of *B. cereus* in milk was faster, reaching 8.73 log_10_ CFU/mL at the end of the 48-h period (Figure 1F). The decrease in *B. cereus* in milk exhibited a positive correlation with the dosage of SE. Compared with the control group, SE at 1× MIC significantly reduced the growth of *B. cereus* in milk by2.54, 2.71 and 2.85 log_10_ CFU/mL at 12, 24 and 48 h, respectively (Figure 1F). No studies have been reported about the effect of SE or sesame oil on the growth of *B. cereus*.

From 0 h to 48 h, the concentration of *S.* Enteritidis in SSS increased from 6.0 log_10_ CFU/mL to 7.76 log_10_ CFU/mL (Figure 1G). Compared with the control group, SE at 1/2× MIC and 1× MIC significantly inhibited the growth of *S.* Enteritidis in SSS within 48 h (Figure 1G). Among these, SE at 1× MIC showed the greatest effect after 24 h with a 2.16 log_10_ CFU/mL reduction (Figure 1G). The concentration of *S.* Enteritidis in milk increased to a maximum of 8.76 log_10_ CFU/mL at 24 h and then gradually decreased thereafter (Figure 1H). For *S.* Enteritidis, the effect of SE was concentration-dependent in milk. Compared with the control group, SE at 1× MIC significantly reduced the growth of *S.* Enteritidis in milk by 2.65, 2.85, and 1.16 log_10_ CFU/mL at 12, 24, and 48 h, respectively (Figure 1H).

The antimicrobial effect of tahini or SE against other serotypes of *Salmonella* in food products was studied previously. Meatballs treated with SE (0.3 and 0.5 g/kg) exhibited a significant reduction in *S*. Typhimurium (ATCC 14028) count by 2.66 and 5.26 log_10_ CFU/g, respectively, after being stored for 3 days at 3 ± 1 °C, when compared with the control 8.79 log_10_ CFU/g [21]. A cocktail of three serotypes of *Salmonella* (*S.* Typhimurium ATCC 14028, *S*. Newport ATCC 6962 and *S.* Montevideo ATCC 5747) was inoculated with 5.6 log_10_ CFU/g into tahini and stored for 16 weeks at 22 and 4 °C, resulting in a decrease of approximately 4.5 and 3.3 log_10_ CFU/g, respectively [35].

The concentration of *E. coli* in SSS increased to a maximum of 7.95 log_10_ CFU/mL at 24 h and then gradually decreased thereafter (Figure 1I). The inhibitory effect of SE against *E. coli* was concentration-dependent both in SSS and milk. Compared with the control group, SE at 1× MIC significantly reduced the growth of *E. coli* in SSS by 1.76, 2.96, and 2.55 log_10_ CFU/mL at 12, 24, and 48 h, respectively (Figure 1I). Similar to the trend in SSS, the *E. coli* in milk increased to a maximum of 8.11 log_10_ CFU/mL at 24 h and then gradually decreased thereafter (Figure 1J). Compared with the control group, SE at 1× MIC significantly reduced the levels of *E. coli* in milk by 3.08, 2.21, and 2.35 log_10_ CFU/mL at 12, 24, and 48 h respectively (Figure 1J). Sesame oil was reported to have growth inhibitory effect on *E. coli* based on the zone of inhibition assay [33]. The viability of *E. coli* O157:H7 and *Listeria innocua* in tahini decreased rapidly when stored at 37 °C, followed by storage at 21 °C and then at 10 °C over a duration of 28 days [36].

The optimal antibacterial effect for different strains is not limited to 48 h, with some exhibiting peak bactericidal activity at 12 or 24 h (Figure 1). In SSS, the order of the most significant reduction in the concentration of five bacteria by SE (1× MIC) was as follows: *L. monocytogenes* (3.63 log_10_ CFU/mL at 24 h) > *S. aureus* (3.29 log_10_ CFU/mL at 12 h) > *E. coli* (2.91 log_10_ CFU/mL at 24 h) > *B. cereus* (2.79 log_10_ CFU/mL at 24 h) > *S.* Enteritidis (2.16 log_10_ CFU/mL at 24 h). However, in milk, the order was *L. monocytogenes* (4.16 log_10_ CFU/mL at 48 h) > *E. coli* (3.08log_10_ CFU/mL at 12 h) > *B. cereus* (2.85 log_10_ CFU/mL at 48 h) = *S.* Enteritidis (2.85 log_10_ CFU/mL at 24 h) > *S. aureus* (2.62 log_10_ CFU/mL at 12 h).

The inhibitory effect of SE on different foodborne pathogens was inconsistent. This study demonstrated that the effectiveness of SE against *L. monocytogenes* was slightly higher compared with the other strains. SE (0.5 g/kg) has been reported to cause significant reductions in *E. coli* O157:H7 (ATCC 51659), *S.* Typhimurium (ATCC 14028), *S. aureus* (ATCC 6538), and *L. monocytogenes* (ATCC 51774) artificially inoculated into meatballs, with decreases on the third day of 5.25, 5.26, 5.1, and 5.17 log_10_ CFU/mL, respectively [21]. However, this report also showed that the growth of *E. coli* O157:H7, *S.* Typhimurium, *L. monocytogenes*, and *S. aureus* in meatballs was completely inhibited by SE treatment (0.5 g/kg) on days 6, 6, 9, and 12 of storage during cold storage at 3 ± 1 °C [21]. SE at a concentration of 2 mg/mL inhibits over 90% of the growth of both *B. cereus* and *S. aureus*, with a clearer zone of inhibition observed against *B. cereus* [27]. Furthermore, the inhibitory activity of sesame oil, which contains 17.91% SE, was observed to be predominant against *E. coli* (27.3 mm) and *S. aureus* (25 mm), compared with that against *Bacillus subtilis* (18.7 mm) and Pseudomonas aeruginosa (12.7 mm), based on the inhibition zone method [33]. Another report showed that sesame oil was more effective to *S*. Typhimurium (NCIM 2493, 25 mm) than that of *S. aureus* (NCIM2602, 19 mm), *B. subtilis* (NCIM2480, 18 mm), and *E. coli* (NCIM2981, 15 mm) [35,37]. Our findings do not completely align with previous reports regarding the order of strain sensitivity to SE, which may be attributed to the utilization of different strains in various studies. The sensitivity to SE has been shown to vary among strains from different sources or with different ST types in our previous studies [20].

Furthermore, we observed a slightly stronger antibacterial effect on *L. monocytogenes*, *S.* Enteritidis, and *E. coli* in pasteurized milk compared with that in SSS. This difference might be attributed to the molecular structure of SE, which includes a phenol group and benzodioxole group, resulting in a higher solubility of SE in the lipid phase [16]. Our previous research showed that the SE showed a better inhibitory effect on *L. monocytogenes* in pasteurized milk compared with that in TSB-YE [19]. Therefore, we hypothesize that pasteurized milk containing a large number of granular fat globules is more conducive to the dispersion of SE and enhances its penetration and antimicrobial efficacy.

### 3.3. Effect of SE on Cell Membrane Permeability of Five Bacterial Strains

The EC of the bacterial suspension could reflect changes in cell membrane permeability [19,38]. As shown in Figure 2, the EC values of the five pathogen strains in the control group remained at approximately 1.21–1.45 mS/cm from 0 h to 48 h. While the EC value of the bacteria suspensions treated with SE showed an increasing trend compared with the control group, the increase in the EC value was higher in the SE group treated with 2× MIC compared with that treated with 1× MIC, but this difference was not statistically significant. Compared with the control group, the EC values of *L. monocytogenes*, *S. aureus*, *B. cereus*, *S.* Enteritidis, and *E. coli* suspensions treated with SE (2× MIC) increased significantly by 0.15, 0.15, 0.13, 0.11, and 0.05 mS/cm, respectively, at 48 h (Figure 2).

These results indicate that SE could cause an increase in the permeability of the cell membranes of the five pathogenic bacteria. The bacterial cytoplasmic membrane serves as a permeable interface for small ions such as Na^+^, K^+^, and H^+^, which is crucial in supporting the activities of the cell membrane, maintaining enzymatic efficacy, and promoting normal metabolic processes [39]. Several studies have shown that the use of antibacterial agents, such as nisin [19], limonene [38], bifidocin A [40], and clove oil [41], can disrupt the cell membranes of bacteria, causing internal electrolytes to leak into the culture medium and subsequently increasing its conductivity. Thus, it is speculated that SE might interact with cellular membranes, resulting in the release of inorganic salts and small ions (such as K^+^, H^+^, and Na^+^), which could potentially serve as a mechanism underlying the bactericidal effect induced by SE.

### 3.4. Effect of SE on Bacterial DNA Leakage

DNA can leak outside the cell when the cellular structure of the bacteria is disrupted [19,41]. Absorbance measurements at 260 nm for DNA content have been considered one of the indicators for assessing cell membrane integrity [42]. The effect of SE on the DNA leakage of all five pathogenic bacteria was analyzed (Figure 3). The DNA content of the five pathogenic bacteria suspensions treated with SE at both 1× MIC and 2× MIC increased within 12–48 h compared with the control group (Figure 3). There was no significant difference in the increase in DNA content between 1× MIC and 2× MIC of SE for *L. monocytogenes*, *S. aureus*, and *E. coli* within 0–24 h (Figure 3A,B,E). However, after 48 h, the DNA content of bacterial suspensions treated with SE (2× MIC) showed a significantly greater increase compared with those treated with 1× MIC for all five strains (Figure 3). Some studies have shown that essential oils (clove oil and essential oil) and phenols (carvacrol and thymol) could cause damage to bacterial cell membranes, resulting in the leakage of nucleic acids [41,43,44]. Thus, the disruption of the cell membrane, resulting in the leakage of DNA, is likely to constitute the underlying antimicrobial mechanism for SE.

### 3.5. Effect of SE on Leakage of Bacterial Protein

The damage caused by SE to the cell membrane of five bacteria was further explored through the determination of the soluble protein content of bacteria suspensions. The protein concentration in the control group remained at 4.7–5.5 mg/mL within 48 h, indicating no leakage of cytoplasmic contents (Figure 4). After 12 h of SE (1× MIC and 2× MIC) treatment, the protein content was significantly increased for all five bacteria compared with the control group (Figure 4). There was no significant difference in the protein content except for *L. monocytogenes* between 1× MIC and 2× MIC of SE. The extracellular protein concentration remains stable from 12 h to 48 h after SE treatment, indicating that most of the proteins inside the cell have leaked out by 12 h. Some similar reports revealed that phenolic acids [45], plant flavonoids [46,47] resulted in the loss of intracellular protein. Therefore, the findings indicate that the application of SE treatment led to substantial harm to the integrity and permeability of the bacterial cell membrane, consequently leading to the release of nucleic acids and proteins from cells.

### 3.6. Effect of SE on ATP Content

ATP plays an important role in bacterial energy metabolism [38]. The effect of SE on the ATP content of the five pathogenic bacteria suspensions was investigated, as shown in Figure 5. The control samples maintained a tendency towards a slight increase in ATP content from 0 h to 24 h, which then stabilized and showed a minor decrease to 48 h (Figure 5). The ATP content of *L. monocytogenes*, *S.* Enteritidis, and *E. coli* suspensions treated with SE (1× MIC) for 24 h was significantly reduced compared with the control group (Figure 5A,D,E), and the trend of significant reduction in *S.* Enteritidis and *E. coli* was maintained until 48 h (Figure 5D,E). The ATP content of *S. aureus* and *B. cereus* suspensions after SE (1× MIC) treatment decreased significantly at 12 h and 48 h compared with the control group, but their ATP content at 24 h was not significantly different from the control group (Figure 5B,C). Compared with the control group, five pathogens treated with SE (2× MIC) all showed a significant reduction in the ATP content within 48 h (Figure 5).

The destabilization of the cytoplasmic membrane or leakage of ions may affect the membrane-associated energy-transducing system [14]. The ATP level serves as an indicator for evaluating the integrity of the cell membrane [48,49]. The exposure of *B. cereus* to 2 mmol/L carvacrol leads to a rapid depletion of intracellular ATP concentration and a slight increase in the extracellular ATP levels within 10 min [50]. The phenolic compound eugenol, upon reaching a concentration of 200× MIC, caused *E. coli* to release 2800 nmol/L intracellular ATP into the medium [49]. The results of our study demonstrated a significant reduction in ATP levels, which may be directly proportional to the decrease in intracellular ATP concentrations. It has been reported that the phenol hydroxyl group of phenolic compounds, such as flavonoids and eugenol, could form hydrogen bond with the active site of the enzyme and inhibit its catalytic activity [15,47]. We speculated that SE might affect the activity of ATP-associated enzymes, leading to a decrease in ATP concentration. In the future, it is necessary to investigate the role of SE in ATPase function and important energy metabolism pathways.

### 3.7. Effect of SE on Bacterial Morphology of Bacterial

A scanning electron microscopy (SEM) analysis was performed to understand the morphological changes in five foodborne pathogenic bacterial strains after exposure to SE. The untreated cells of the five strains appeared morphologically normal, intact, and had smooth surfaces without any changes in morphology (Figure 6). *S. aureus* and *B. cereus* treated with 1× MIC of SE showed different degrees of cell-to-cell adhesion, bacterial volume expansion, unclear cell wall boundaries, and relatively flat morphology compared with the control group. The *L. monocytogenes*, *S. enteritidis*, and *E. coli* treated with SE (1× MIC) exhibited more pronounced morphological changes, including partial wrinkling, collapse, surface fracture, and deformation. Compared with 1×MIC, treatment with 2× MIC of SE resulted in significant bacterial cell damage and breakdown of cellular structure. *B. cereus* treated with 2× MIC of SE exhibited reduced cell size and a collapsed surface, but no complete cell fragmentation was observed. In contrast, Gram-negative *S.* Enteritidis and *E. coli* experienced complete cell breakdown (Figure 6).

Previous studies have also reported that natural phenols can alter the morphology of foodborne pathogens, leading to cell wall separation, cell deformation, and even lysis. It is reported that the application of carvacrol resulted in the release of intracellular contents and the detachment of cell walls, ultimately leading to cell lysis in *L. monocytogenes* cells via transmission electron microscopy analysis [44]. The treatment with eugenol resulted in a change in the significant morphology of *S. enteritidis* and *E. coli* [39,49]. Some phenolic extracts, such as crude hibiscus sabdariffa phenolic-rich extracts and *Spirulina* polyphenolic compounds, lead to cell deformations, wrinkles, and the loss of *S. aureus* cell shapes based on SEM analysis [51,52]. The SEM analysis provided additional evidence that SE effectively suppressed bacterial growth, which was associated with cellular structural impairment. In addition, the difference in susceptibility to phenols between Gram-positive and Gram-negative bacteria is a controversial issue. The antibacterial efficacy of phenols is generally more pronounced against Gram-positive bacteria compared with Gram-negative ones, primarily due to the presence of a lipopolysaccharide layer in the latter, which restricts the penetration and dissolution of extract compounds on their cell membrane [53]. SEM analysis (Figure 6) showed that Gram-negative bacteria (*E. coli* and *S. enteritidis*) treated with 2× MIC of SE displayed more obvious cell lysis compared with the three Gram-positive strains. However, the time–kill assay (Figure 1) showed that *L. monocytogenes* and *E. coli* were more sensitive to SE. Thus, it is speculated that the bacteriostatic mechanism of the SE may differ between Gram-negative and Gram-positive bacteria.

## 4. Conclusions

SE exhibited antimicrobial activity against five different foodborne pathogens. Under different culture conditions (SSS and pasteurized milk), SE inhibited the growth of foodborne pathogens by interfering with the function and morphology of bacterial cells. The antibacterial mechanism of SE might be associated with an increase in cell membrane permeability, the leakage of intracellular substances, and the disruption of cellular structures. The antibacterial activity of SE against foodborne pathogens makes it a promising strategy for future use in food, replacing synthetic preservatives and antibiotics. The future research will aim to further elucidate the bacteriostatic mechanism of SE, such as its specific site of action on bacteria and its interaction with lipid bilayers and cell membranes. In order to enhance comprehension of the bactericidal activity of SE and its impact on food shelf life, it is important to broaden the scope of food matrices and include a wider variety of foodborne pathogens. Furthermore, future research will focus on elucidating the impact of environmental factors (such as heat, cold, acid, and salt) on the antibacterial activity of SE in various food applications.

## Figures and Tables

**Figure 1 foods-13-00435-f001:**
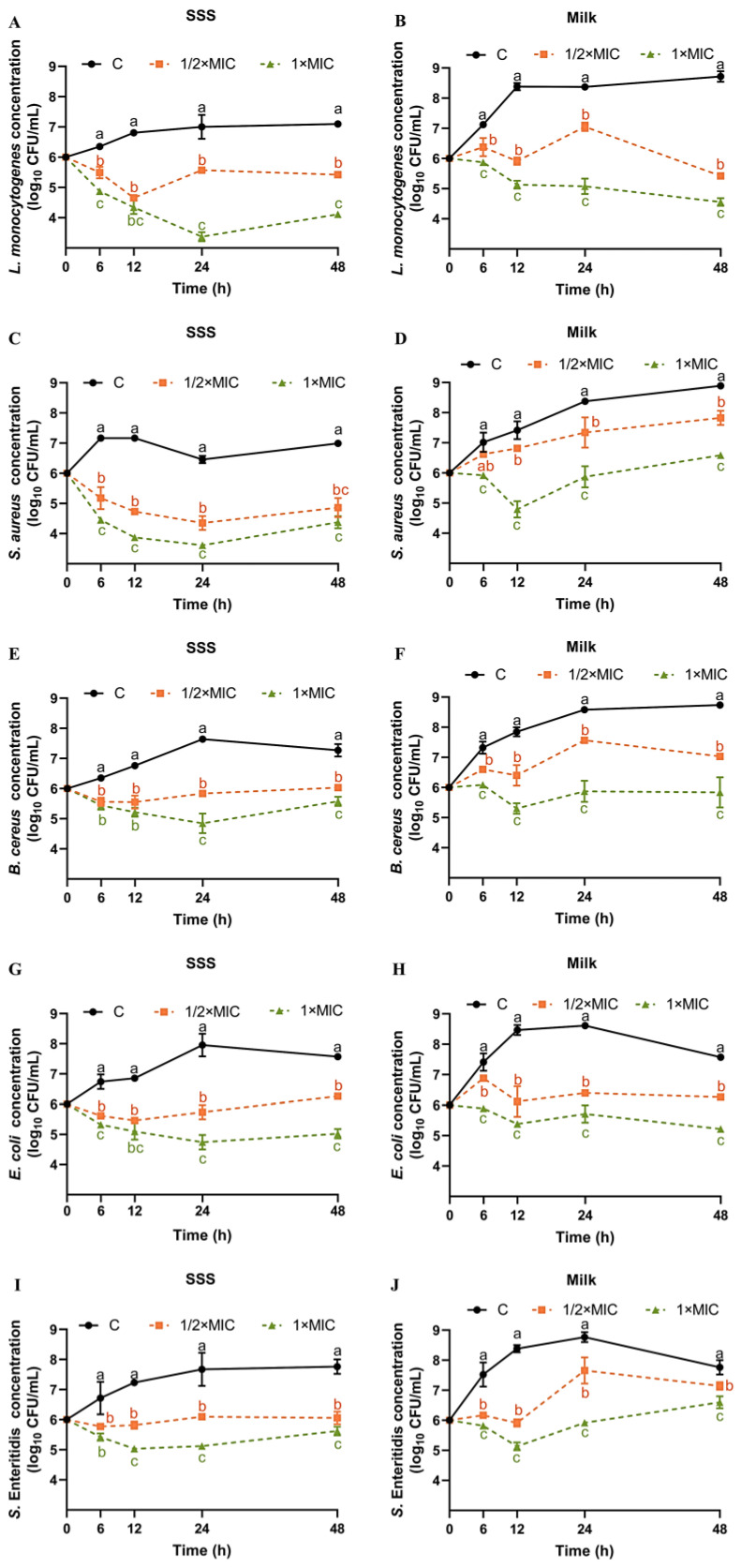
Time–kill curves for SE against *L. monocytogenes* (**A**,**B**), *S. aureus* (**C**,**D**), *B. cereus* (**E**,**F**), *S.* Enteritidis (**G**,**H**), and *E. coli* (**I**,**J**) in SSS and pasteurized milk. Bacteria at a starting inoculum of 10^6^ CFU/mL were treated with or without SE (1/2× MIC and 1× MIC) for 0, 6, 12, 24, and 48 h at 37 °C. (●) C: Control, (■) SE (1/2× MIC), and (▲) SE (1× MIC). Different lowercase letters (a, b, and c) indicate significant differences (*p* < 0.05) among different groups at the same time point.

**Figure 2 foods-13-00435-f002:**
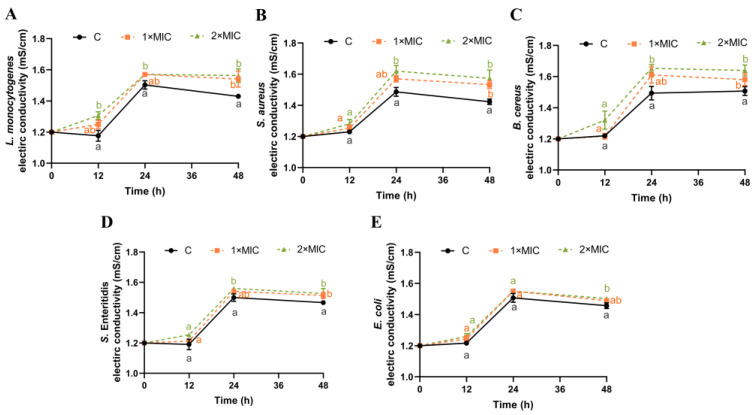
Effect of SE on bacterial membrane conductivity. *L. monocytogenes* (**A**), *S. aureus* (**B**), *B. cereus* (**C**), *S.* Enteritidis (**D**), and *E. coli* (**E**) were used in experiments. The bacteria at a starting inoculum of 10^8^–10^9^ CFU/mL were treated with or without SE (1× MIC and 2× MIC) for 0, 12, 24, and 48 h at 37 °C. Different lowercase letters (a, b, and c) indicate significant differences (*p* < 0.05) between treatments at each time point.

**Figure 3 foods-13-00435-f003:**
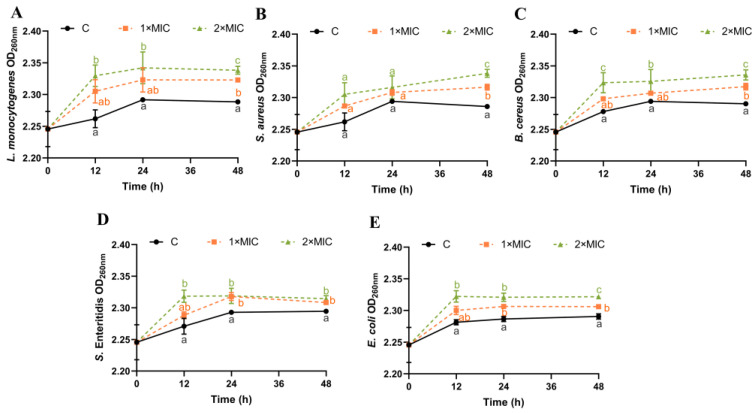
Changes in bacterial OD_260nm_. *L. monocytogenes* (**A**), *S. aureus* (**B**), *B. cereus* (**C**), *S.* Enteritidis (**D**), and *E. coli* (**E**) were used in experiments. The bacteria at a starting inoculum of 10^8^–10^9^ CFU/mL were treated with or without SE (1× MIC and 2× MIC) for 0, 12, 24, and 48 h at 37 °C. Different lowercase letters (a, b, and c) indicate significant differences (*p* < 0.05) between treatments at each time point.

**Figure 4 foods-13-00435-f004:**
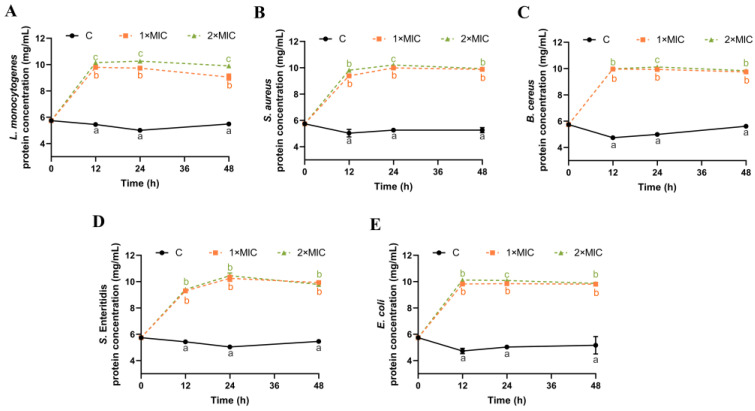
The effect of SE on the concentration of leaked proteins from *L. monocytogenes* (**A**), *S. aureus* (**B**), *B. cereus* (**C**), *S.* Enteritidis (**D**), and *E. coli* (**E**). The bacteria at a starting inoculum of 10^8^–10^9^ CFU/mL were treated with or without SE (1× MIC and 2× MIC) for 0, 12, 24, and 48 h at 37 °C. Different lowercase letters (a, b, and c) indicate significant differences (*p* < 0.05) between treatments at each time point.

**Figure 5 foods-13-00435-f005:**
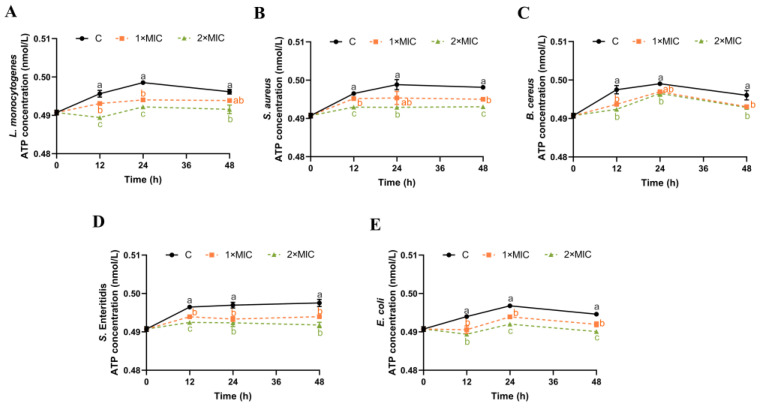
The effect of SE on the ATP content in *L. monocytogenes* (**A**), *S. aureus* (**B**), *B. cereus* (**C**), *S.* Enteritidis (**D**), and *E. coli* (**E**). The bacteria at a starting inoculum of 10^8^–10^9^ CFU/mL were treated with or without SE (1× MIC and 2× MIC) for 0, 12, 24, and 48 h at 37 °C. Different lowercase letters (a, b, and c) indicate significant differences (*p* < 0.05) between treatments at each time point.

**Figure 6 foods-13-00435-f006:**
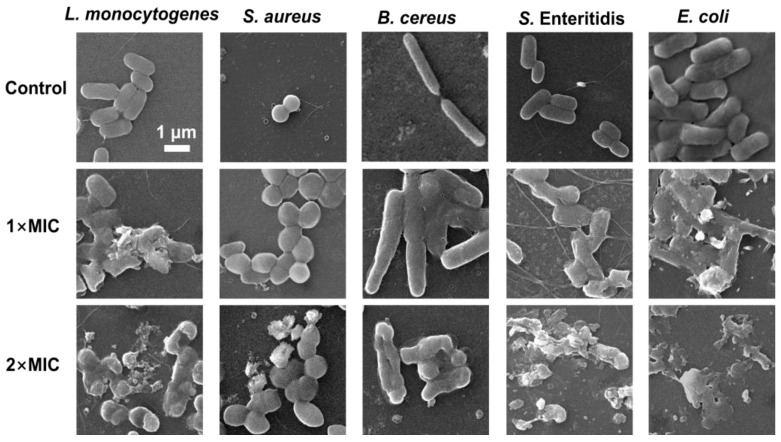
SEM observation of *L. monocytogenes*, *S. aureus*, *B. cereus*, *S.* Enteritidis, and *E. coli* exposed to SE. The bacteria at a starting inoculum of 10^6^ CFU/mL were treated with or without SE (1× MIC and 2× MIC) for 24 h at 37 °C.

**Table 1 foods-13-00435-t001:** MICs and MBCs of SE against different foodborne strains.

Foodborne Pathogens	Strains	SE (mg/mL)			Penicillin (μg/mL)
		MIC in SSS	MIC in BHI	MBC in SSS	
*L. monocytogenes*	ATCC 19112	2.0	2.0	4.0	16
*S. aureus*	ATCC 29213	1.5	2.0	16.0	4
*B. cereus*	ATCC 14579	2.0	2.0	4.0	16
*S.* Enteritidis	MRL 04120	1.5	1.5	8.0	128
*E. coli*	ATCC 25922	1.5	1.5	4.0	128

MRL: the abbreviation of the author’s laboratory, “Microrisk Lab”; ATCC: American Type Culture Collection; MIC: minimum inhibitory concentration; MBC: minimum bactericidal concentration; SE: sesamol; SSS: 0.85% sterile saline solution (pH = 7.2); BHI: brain heart infusion broth.

## Data Availability

All data related to the research are presented in the article.

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
