# Peer review of "Antibacterial Effect and Possible Mechanism of Sesamol against Foodborne Pathogens"

_foods, 2024, doi:10.3390/foods13030435_

Round 1

Reviewer 1 Report

Comments and Suggestions for Authors

This research investigates the antibacterial properties of Sesamol (SE), a phenolic compound found in sesame, against various foodborne pathogens. The abstract highlights the global concern over food safety due to foodborne pathogens and the need for effective bacteriostatic agents. The study aims to evaluate SE's antibacterial effect and potential mechanisms against pathogens such as Listeria monocytogenes, Staphylococcus aureus, Bacillus cereus, Escherichia coli, and Salmonella serovar Enteritidis. Results demonstrate significant inhibition of bacterial growth in saline and pasteurized milk. Notably, L. monocytogenes proves more sensitive to SE. The study explores alterations in cell membranes, release of intracellular components, and changes in extracellular ATP concentration as potential mechanisms. The scanning electron microscope analysis reveals notable morphological changes in bacteria exposed to SE, suggesting its effectiveness against both gram-positive and gram-negative bacteria.

Positive Aspects:

The article effectively addresses a critical issue—food safety—and proposes Sesamol as a potential natural antimicrobial agent. The introduction succinctly outlines the relevance of foodborne pathogens, emphasizing the severity of associated illnesses. The comprehensive exploration of SE's mechanisms and effects on various bacteria adds depth to the study. Notably, the inclusion of both gram-positive and gram-negative strains enhances the applicability of the findings. The material and methods section provides clarity on experimental procedures and bacterial strains.

Improvement Suggestions:

  1. Clarify the language in the abstract for a smoother flow and understanding.
  2. Refine sentences in the introduction to enhance conciseness and clarity.
  3. Simplify certain technical details in the materials and methods section for accessibility.
  4. Ensure consistent formatting and style throughout the article for a polished presentation.

Author Response

We have provided a point-to-point response to your question in the attachment. Thank you for your helpful feedback on our article.

Reviewer 2 Report

Comments and Suggestions for Authors

1. Please work on editing your manuscript. There are a number of minor but significant errors, e.g. Latin names should be written in italics, Gram-positive and Gran-negative (not gram positive and gram negative) are correct, etc.

2. MIC determination should be carried out in culture medium, according to MIC determination standards , not in SSS. I cannot agree with the sentence in lines 182-184.

3. The determined MIC values are very high, in my opinion the tested compound has a very low antimicrobial activity. Please compare the determined values to a standard antibiotic.

4) Is SE not toxic in these high concentrations? Please show the results for toxicity to normal cells.

5. Please show the growth curves (A at 600 nm as a function of time) from which the MIC values were estimated for each test strain. Compare with growth in broth.

6. The bactericidal concentration is determined above the MIC value - this is the MBC value. At concentrations below the MIC, e.g. survival can be determined - I propose the LIVE/DEAD test with Syto-9 and PI probes.

7. I would also suggest measuring permeability and DNA secretion using a fluorescent method, e.g. SYTOX Green.

8. I have doubts about the spectrophotometric measurements because the authors do not show anywhere in the manuscript what colour SE is at what wavelength the maximum absorption occurs for this compound. Doesn't this affect the absorbance measurements of various parameters.

9. Does SE induce structural changes in the membranes of individual strains?

Author Response

(The authors gave the same response as above.)

Reviewer 3 Report

Comments and Suggestions for Authors

The manuscript is interesting and in global terms is well written. Although the effects of sesamol against food pathogens was widely investigated in the past, the manuscript includes some novelty information such as the investigation about the mechanisms of action of the observation of the bacterial species by scanning electron microscope. In my opinion, the two major drawbacks of the manuscript are the following:

1)      In the first part of the manuscript, there are a reiterative comparison of the results obtained with those previously reported for meatballs. I consider that such a comparison is not adequate, since they are very different matrices and that the article should compare its results with those obtained for sesamol in milk, or at least in beverages.

2)      Any work proposing the use of bacterial growth inhibitors as aromatic as sesamol in foods must prove that their use, at the concentrations used, does not cause organoleptic alteration in the food, or at least not a drastic alteration.

Additionally, there are some minor aspects that should be considered:

1)      In some parts of the work, the actions carried out are written in the present tense (examples, lines 14, 274). Normally, these activities are written in the past tense. I recommend that you check the manuscript and write the activities performed in the past tense.

2)      There are some considerations on the use of abbreviations that should be corrected. For example, "MBCs" and "MIC" have not been defined in the abstract. Also, since they are only cited once, it does not make much sense to abbreviate them instead of naming them by their full name. In tables, the table itself should make sense independently of the text, so in Table 1, all abbreviations should be defined in the table footer. Also, "MRL" was not defined in line 95.

Otherwise I think it is a good article

Author Response

(The authors gave the same response as above.)

Round 2

Reviewer 2 Report

Comments and Suggestions for Authors

The authors of the manuscript did not comply with many of my comments.

Please show the growth curves (A at 600 nm as a function of time) from which the MIC values were estimated for each test strain. Compare with growth in broth. The graphs presented in the responses (comments 5) do not respond to my request. Please respond to my question. Such curves should be presented in the manuscript.

Please add in Table 2 the MIC values obtained for the antibiotic at each tested strain. Please do not provide values taken from the literature.

Is SE not toxic in these high concentrations? Please show the results for toxicity to normal cells. You should show your own results obtained for the extract tested, not literature values!!!!!!

Comments 6 and 7 - no specific answer to my question. The authors write about what has gone before.

Author Response

Your suggestions are very helpful for our manuscript. We have provided a point-to-point response to your question in the following document.
